# First Diagnostic Questionnaire for Assessing Patients’ Social Functioning: Comprehensive DDX3X Syndrome Patient Profile

**DOI:** 10.3390/jcm13247842

**Published:** 2024-12-22

**Authors:** Urszula Stefaniak-Preis, Ada Kaczmarek, Mirosław Andrusiewicz, Magdalena Roszak, Natalia Trzeszczyńska, Włodzimierz Samborski, Ewa Mojs, Roksana Malak

**Affiliations:** 1Department of Clinical Psychology, Poznan University of Medical Sciences, Bukowska 70, 60-812 Poznań, Polandewamojs@ump.edu.pl (E.M.); 2Department of Cell Biology, Faculty of Health Sciences, Poznan University of Medical Sciences, Rokietnicka 5D, 60-806 Poznań, Poland; andrus@ump.edu.pl; 3Department of Computer Science and Statistics, Poznan University of Medical Sciences, Rokietnicka 7, 60-806 Poznań, Poland; 4Department and Clinic of Rheumatology, Rehabilitation and Internal Diseases, Poznan University of Medical Sciences, 28 Czerwca 1956 r. 135/147, 61-545 Poznań, Poland

**Keywords:** DEAD-box helicase 3 X-linked gene (DDX3X), autism spectrum disorder (ASD), developmental delay (DD), intellectual disability (ID), ultra-rare disease (URD), neurodevelopmental issues

## Abstract

**Background/Objectives:** DDX3X syndrome is often misdiagnosed as autism spectrum disorder (ASD, Rett Syndrome, and Dandy–Walker Syndrome). Precise phenotyping is needed with reference to neurodevelopmental diagnosis. Observation of behavior and communication in parents with DDX3X syndrome in the USA, France, and Poland; conversations with the parents of patients; and rudimentary information in evidence-based medical articles prompted us to identify differences in communication, play, and social interaction between children with ASD only, those with both ASD and *DDX3X*, and those with *DDX3X* only. **Methods:** As diagnostic tool for *DDX3X* patients, we created a questionnaire divided into four sections: medical, social, play, and communication. **Results:** The results showed inconsistent diagnoses in different countries where children could have been diagnosed with *DDX3X*. In a comparative analysis, individuals with *DDX3X* exhibited greater social skills than individuals with ASD. Furthermore, those with *DDX3X* demonstrated higher levels of social functioning compared to children with ASD. Therefore, parents of children recently diagnosed with ASD or similar conditions are encouraged to complete a survey to determine if their child is likely to have features of DDX3X syndrome. **Conclusion:** Identification of early behavioral markers that differentiate children with ASD and those with *DDX3X* could lead to the earliest opportunity for identification and intervention, and can significantly impact developmental trajectories, leading to better long-term outcomes.

## 1. Introduction

The DEAD-box helicase 3 X-linked gene (*DDX3X*) defect is linked to a developmental disorder, which was first documented in 2015 and is considered to be an ultra-rare congenital genetic condition [1,2,3]. There have been approximately 100 reported variations with a (likely) harmful nature in this gene globally, and the existing literature outlines the progression of the condition in around 150 individuals [4,5]. As the gene is situated on the X chromosome, the majority of the diagnosed patients are females. It is believed that the condition impacts 1–3% of women with unexplained intellectual disability [1,6].

The *DDX3X* gene encodes a highly conserved RNA helicase of the DEAD-box family [7,8]. It involves various essential cellular functions, including transcription, splicing, RNA transport, and translation [1]. Mutations in *DDX3X* are associated with conditions such as viral infections, inflammation, and intellectual and developmental disabilities [9,10]. Additionally, it has been demonstrated that the condition is linked to neurological symptoms, motor delays, behavioral problems, cardiac dysfunction, and ophthalmic and gastrointestinal abnormalities [1,3,11,12]. *DDX3X* variants have been also associated with distinct brain MRI abnormalities and brain tumors [9,13]. The corpus callosum hypoplasia, polymicrogyria, and ventricular enlargement are the most prevalent brain malformations [14]. Observable presentations of psychopathological symptoms encompass modifications in the behavioral, cognitive, and affective domains [5,15,16].

Globally, there are 1180 cases of *DDX3X* patients, comprising 40 males and 1140 females across 56 countries (https://ddx3x.org/, accessed on 7 August 2024). However, 200 children with *DDX3X*-related disorders (*DDX3X*-RD) figure in scientific registers [17]. A thorough examination of cases in conjunction with whole exome sequencing (WES) is essential before confirming a *DDX3X* syndrome diagnosis, as *DDX3X* has previously been reported to mimic cerebral palsy or be diagnosed as ASD [12,16,18]. Referring to the Diagnostic and Statistical Manual of Mental Disorders (DSM-5), autism spectrum disorder (ASD) primarily impacts communication and behavior [19,20]. Currently, the prevalence of ASD is on the rise [21]. The prevalence of autism is 1 in 34. Regrettably, individuals with ASD often do not have access to comprehensive genetic testing. This is extremely important because screening would serve to speed up diagnosis and introduce appropriate therapy during the period of greatest brain plasticity. Additionally, a recent article in *Nature* by Inge Kamp Baker highlighted the significant risk of incorrect positive ASD diagnoses [22].What is more, many parents who came to learn about a child’s diagnosis of ASD can experience a trauma [23,24]. The failure to diagnose ASD can lead to therapeutic approaches not appropriate for a child [25].

Facial dysmorphism is a phenotypic characteristic that frequently prompts parents and caregivers to pursue an alternative diagnosis. The current medical literature on behavior in DDX3X syndrome patients is extremely limited. Although some papers mention behavioral symptoms of DDX3X syndrome [5,6,15,26,27], we were unable to find any articles that discuss the positive social aspects of behavior in individuals with *DDX3X*. Other than two brief references in medical literature, we have not come across any positive social aspects of behavior in *DDX3X* patients [15,16]. First, there is a reported case study of a seven-year-old girl who displayed behaviors similar to autism, such as repetitive behaviors, which improved with behavioral interventions and inclusive education. Some of her traits did not meet the diagnostic criteria for ASD, including cognitive curiosity, spontaneous exploration, and a strong desire for social interaction despite a lack of speech [16]. Second, parents in medical interviews mentioned the strengths and qualities of *DDX3X* females, including caring and friendly personalities, a sense of humor and laughter, love of dancing and swimming, and a willingness to persevere in the face of challenges [15]. A recent review on psychopathological behavior in DDX3X syndrome identified specific psycho-pathological features exhibited by patients [6]. An understanding of the positive behavioral and social profiles could provide valuable diagnostic markers and assist healthcare practitioners in identifying and treating patients earlier. A deeper understanding of behaviors can also drive the development of targeted therapies, behavioral interventions, and supportive tools. Applying practices suitable to the patient’s abilities and strong behavioral traits could improve the quality of life for patients.

Children with DDX3X have many features that are similar to those in children without intellectual disabilities [5]. However, not all children with ASD have intellectual disability [28]. That is why we have decided that we would not consider just “intellectual disability” as the feature that differs the group of children with DDX3X and children with ASD.

The aim of this study was to identify differences among children with only ASD, ASD and *DDX3X*, or just *DDX3X* diagnosis in communication, play, and social interaction domains. The secondary objective was to create a tool for the purpose of assisting in the diagnosis and provision of appropriate therapy for pediatric patients while also facilitating parental decision-making in the selection of suitable early education facilities and educational institutions for their children.

## 2. Materials and Methods

A custom-made questionnaire (Appendix A) was used for the research. It was posted in a group for parents of children with ASD and *DDX3X* and handed out to parents of children with *DDX3X* at the First International Conference on DDX3X Syndrome in Paris, France (20 October 2023). We divided the questionnaire into four parts: medical, social skills, play, and communication. The questionnaire’s medical part was not scored and was based on the clinical characteristics of a given child. Other parts were scored (Appendix A).

The Social Behavior Questionnaire contains 12 items in the medical domain (MD), 14 items in social skills domain (SS), 14 items in the play domain (PD) and 6 items in the communication domain (CD). Questions from the MD and the 7th item of the SS were excluded from scoring, as they did not contribute to the main aim of differentiating patients. We decided to included MD_9 and MD_10 from that category, as we wanted to examinate whether facial dysmorphism and microcephaly might have had an influence on other results in other domains. Parents could answer by matching the sentence that described their child’s behavior the best. Two questionnaires were excluded from the statistics, as they were not fully completed. The scoring system was created based on our team’s observations. We would like to standardize our diagnostic tool.

The questionnaire has been translated into English and French. A total of 112 parents of patients from 3 different continents responded to the questionnaire (parents from Europe: Poland, Greece, Ireland, Ukraine, Hungary, the UK, Spain, Germany, Belgium, Norway, France, and Italy; parents from Asia: Japan, Turkey, India, and Russia; as well as parents from North America and Canada). There was a total of 112 children, including 42 patients with ASD (14 females and 28 males) and 70 patients with *DDX3X* (4 males and 66 females). Out of the 70 *DDX3X* patients, 49 were diagnosed with *DDX3X* and 21 were diagnosed with both ASD and *DDX3X*). Out of the female patients, 15 had ASD, 47 had *DDX3X*, and 20 were diagnosed with both (Figure 1). In the group of male patients, 27 had ASD, 2 had *DDX3X*, and 2 were diagnosed with both disorders. The oldest patient was 41 years old and the youngest was 2. On average, the study subjects were 10 years old.

### 2.1. Statistical Analysis

Statistica version 13.3 (TIBCO Software Inc., Palo Alto, CA, USA) and PQStat 1.8.0.414 (PQStat software; Poznań, Poland) were used for statistical analyses. JMP Pro v.17.0.0 (622753) software (SAS Institute, Cary, NC, USA) was used for graphical data visualizations.

Nominal data distributions (2 × 2 tables) for different groups and subgroups were compared using chi-square tests according to Cochran’s rule, which also provided the odds ratio along with the confidence interval for significant relationships. For more than one degree of freedom, the Pearson’s chi-square test was used if Cochran’s conditions were met. If Cochran’s conditions were not met, Fisher’s exact test for CxR tables was used. The Bonferroni–Hochberg test was the post hoc test of choice. Results are presented in terms of frequencies. The Cramér’s phi coefficient was used to measure associations (φ_c_) [29,30].

The Shapiro–Wilk test was used to test quantitative variable distributions for normality. As the null hypothesis was rejected for the distribution of variables, the Mann–Whitney *U* test with corrections was used to determine differences. The *Z*-score was used to measure associations [30]. The median (*Me*) and interquartile range (Q1–Q3) were used to describe the results.

### 2.2. Reliability 

A reliability analysis was carried out using the Kuder–Richardson 20 method because the data are dichotomous (0/1), excluding two questions in the communication scale (2 and 8) because it has more categories.

## 3. Results

The study comprised thirteen questions in the social skills domain (SS_1–SS_6, SS_8–SS_14; question SS_7 was excluded from the statistical analysis), fourteen questions in the play domain (PD_1–PD_14), and six questions in the communication domain (CD_1–CD_6). The children were divided into two groups: those diagnosed with autism spectrum disorder and those with DDX3X syndrome (due to the rarity of the syndrome, children diagnosed with *DDX3X* mutations with and without ASD were included in this group). Reliability for the entire tool was satisfactory (0.738). The statistical power was determined posteriori following data collection. As the percentages of ASD and DDX3X children were 62.5% to 37.5% respectively, the achieved statistical power was 0.734 for the group of 112 children.

### 3.1. Social Behavior of ASD & DDX3X Children

There were significant differences for 6 out of 13 questions analyzed in the social skills domain, 9 out of 14 questions in the play domain, and 1 out of 6 questions analyzed in the communication domain between children with ASD and DDX3X syndrome (Figure 2).

#### 3.1.1. Social Skills

For SS_1 (“happy child”), 23% of the ASD group and 33% of the *DDX3X* group were in the “No” category, with 76% and 67%, respectively, in the “Yes” category (*p* = 0.03924; Figure 2, Appendix A). For SS_2 (“sociable with friends, family, and people”), 31% of the ASD group and 10% of the *DDX3X* group were in the “No” category, with 69% and 90%, respectively, in the “Yes” category (*p* = 0.00577). For SS_4 (“good disposition, caring and friendly personality”), 12% of the ASD group and 3% of the *DDX3X* group were in the “No” category, with 88% and 97%, respectively, in the “Yes” category (*p* = 0.00256). For SS_5 (“powerful urge to make friends”), 83% of the ASD group and 37% of the *DDX3X* group were in the “No” category, with 17% and 63%, respectively, in the “Yes” category (*p* = 0.00001). For SS_6 (“developing and maintaining friendships”), 76% of the ASD group and 51% of the *DDX3X* group were in the “No” category, with 24% and 50%, respectively, in the “Yes” category (*p* = 0.00939). For SS_8 (“does your child recognize close and distant friends”), 23% of the ASD group and 33% of the *DDX3X* group were in the “No” category, with 76% and 100%, respectively, in the “Yes” category (*p* = 0.00003).

#### 3.1.2. Play Domain

Considering responses that differed significantly in the play domain, the majority were in the “Yes” category, similar to the trend for PD_4 (“love listening to music”). Differences between children with ASD and *DDX3X* were found for the PD_2–3, PD5–7, PD_9–10, and PD_13–14 categories (Figure 2; Appendix A). There was a particular difference in questions PD_2 (“fascination with water”), (*p* = 0.02830); PD_3 (“love of swimming”), (*p* = 0.01473); and PD_5 (“love of dancing”), with “Yes” for 64% with ASD and 83% with *DDX3X* (*p* = 0.02611); in PD_6 (“sitting together, playing together, family entertainment together”), (*p* = 0.00604); PD_7 (“is your child interested in what other children are doing, is your child playing, watching or imitating”; *p* = 0.02240); PD_9 (“does your child join in play without knowing rules of the game”; *p* = 0.00015); and PD_10 (“is your child interested in other children, does your child smile at them, approach them”), with ASD 60% and 81% *DDX3X* (*p* = 0.01127); and in PD_13 (“spontaneously initiating conversation/play”; *p* = 0.01777) and PD_14 (“exceptional memory for details”; *p* = 0.00588). After correcting for multiple comparisons, all significant categories in the PD domain remained significant (*p* < 0.05).

#### 3.1.3. Communication Domain

Analyzing the communication domain, one difference with the majority of responses in the “Yes” category was found. Differences between children with ASD and *DDX3X* were found for the CD_5 category (“testing a parent’s reaction”; Figure 2; Appendix A). A total of 64% of the ASD group and 41% of the *DDX3X* group were in the “No” category, with 36% and 59%, respectively, in the “Yes” category (*p* = 0.01917). After correcting for multiple comparisons, CD_5 lost its significance (Appendix A). There were no significant differences in the other categories (*p* > 0.05).

To summarize, the social behavior of children with ASD and those with DDX3X seemed to differ in both groups of children in terms of features such as the play domain, communication domain, and social skills domain.

### 3.2. Facial Dysmorphia

Considering the only question regarding the medical domain (MD_9—facial dysmorphia) with “Yes” responses, one significant difference in each of the domains analyzed was found (Figure 2). For the social skills domain, there was a difference in the SS_3 (“does your child often smile, laugh, does your child have a sense of humor”) category, with 67% of the ASD group and no one from the DDX3X syndrome group in the “No” category, and 33% and 100%, respectively, in the “Yes” category (*p* = 0.00147; Appendix A). Question SS_3 (“does your child smile often, does your child have a sense of humor”) was significant for *p* < 0.05. For the play domain, there was a difference in the PD_4 (“loves listening to music”) category (Figure 1, Appendix A), with 33% of the ASD group and no one from the DDX3X syndrome group in the “No” category, and with 67% and 100%, respectively, in the “Yes” category (*p* = 0.00571).

For the communication domain, there was a difference in CD_3 (“problems in expressing needs, desire to be understood”; Figure 2, Appendix A), with “Yes” declared by 33% in the ASD group and by 91% in the *DDX3X* group (*p* = 0.02912).

To summarize, facial dysmorphia differentiated the two groups of patients, but not all patients presented this feature. Furthermore, the number of subjects with dysmorphia in both groups was too small to compare patient functioning in the context of this feature.

### 3.3. ASD & DDX3X Behavior Across Languages

There were significant differences for 6 out of 13 categories analyzed in the social skills domain, 4 out of 14 questions in the play domain, and none in the communication domain between children in various language groups regardless of whether they were diagnosed with ASD or DDX3X syndrome (Figure 3a,b; Appendix A). In the social skills domain, differences were found for SS_2-SS_8 (2—“sociability with family and people”, 3—“frequent smiling, humor,” 4—“good disposition, friendly personality,” 5—“powerful urge to make friends,” 6—“developing and maintaining friendships,” 8—“making acquaintances and friends”; *p* < 0.05; Appendix A). For the SS_2 (“sociable with family, friends) and the SS_4 (“good disposition, friendly personality”) categories, there was a difference between PL and EN (*p* = 0.0088 and *p* = 0.0003, respectively), and for the SS_3 (“frequent smiling, sense of humor”) category, there were differences between EN and PL (*p* = 0.0459) and between EN and FR (*p* = 0.0466). For the next three scales, SS_5–SS_8 (5—“strong desire to make friends,” 6—“developing and maintaining friendships,” 8—“getting to know acquaintances and friends”), differences were found between PL and EN (*p* = 0.0285, *p* = 0.0095, and *p* = 0.0001, respectively) and between PL and FR (*p* = 0.0018, *p* = 0.0154, and *p* = 0.0117, respectively). After correcting for multiple comparisons, SS_2–3 (2—“sociability with family and people,” 3—“frequent smiling”) and SS_6 (“developing and maintaining friendships”) could only be said to be trends. In the play domain, there were differences in PD_1 (“spontaneous exploration of the environment”), PD_6 (“sitting together at a table”) and PD_13-14 (Appendix A; 13—“initiation of play or conversation,” 14—“exceptional memory for details”). For PD_1 (“spontaneous exploration”), there were differences between PL and FR (*p* = 0.0022) and between FR and EN (*p* = 0.0001). For PD_6 (“sitting together at a table”) and PD_14 (“exceptional memory for details”), there was a difference between PL and EN (*p* = 0.0107 and *p* = 0.0008, respectively). For PD_13 (“initiation of spontaneous play”), differences were found between EN and PL (*p* = 0.0146) and between EN and FR (*p* = 0.0017). There were no differences observed in the communication domain (*p* > 0.05, Figure 3a; Appendix A).

For children with ASD, two significant differences were found between language groups in the social skills domain (Figure 3a,b; Appendix A). There was a difference in the SS_3 category (“frequent smile, sense of humor”) (*p* = 0.00674) between the FR and EN groups (*p* = 0.0038) and in the SS_6 category (“developing and maintaining friendships”; *p* = 0.04768). In this domain, there was also a trend for SS_8 (“meeting acquaintances and friends”) (*p* = 0.05277). After correcting for multiple comparisons, the level of significance changed into a trend (*p* = 0.08772) only in the SS_3 (“smile, sense of humor”) domain. Among children with ASD, the differences between language groups in the play domain were analyzed for PD_2 (“fascination with water”; *p* = 0.04768) and PD_8 (*p* = 0.00309) between PL and FR (*p* = 0.0826, trend) and between PL and EN (*p* = 0.0107) (Figure 3a,b, Appendix A). There was a significant difference between the language groups for PD_8 (“cognitive aptitude”) (*p* = 0.04326). Two differences were found in the communication domain (Figure 3a,b, Appendix A). There were differences in the CD_1 (“does your child speak”; *p* = 0.01374) and CD_6 (“finger pointing”; *p* = 0.02565) categories between PL and EN (*p* = 0.0309) and between PL and FR (*p* = 0.0409).

For children with DDX3X syndrome, differences were found for question SS_9 between language groups in the social skills domain (Figure 3a,b, Appendix A) (*p* = 0.01185). In the group of children with DDX3X syndrome and the analyzed language group differences in the play domain, significant differences were found in PD_1 (*p* = 0.0004), PD_13 (“initiates play”; *p* = 0.00261), and PD_14 (“exceptional memory”; *p* = 0.03316). For PD_1 (“spontaneous exploration of the environment”), there were differences between PL and FR (*p* = 0.0019) and between FR and EN (*p* = 0.0001). For PD_13 (“initiating spontaneous play with a parent”) and PD_14 (“memory for details”), differences were found between FR and EN (*p* = 0.0026 and *p* = 0.0085, respectively; Figure 3a,b, Appendix A). In the analysis of multiple comparisons, specific differences were found for questions PD_1 (“spontaneous exploration of the environment”) and PD_13 (“spontaneous play with a parent”), with *p* = 0.00056 and *p* = 0.01827, respectively.

To summarize, no significant differences were identified in any language category in the communication domain (Figure 3a, Appendix A).

### 3.4. Age & Skills in ASD & DDX3X

There were significant differences for 1 out of 13 categories analyzed in the social skills domain and 2 out of 14 questions in the play domain, and 1 difference in the communication domain for age regardless of whether they were diagnosed with ASD or DDX3X syndrome (Figure 4). In the social skills domain, the difference was for SS_14 (“overreacting or underreacting in social situations”; *p* = 0.03437; Appendix A). The median age for “Yes” children was significantly higher compared to “No” children (*Me* = 10 [6,7,8,9,10,11,12,13] and *Me* = 7 [5,6,7,8,9,10,11], respectively). A trend was also found for SS_5 (“strong desire to make friends”; *p* = 0.05721) that increased with the age of the children in the “Yes” category (*Me* = 9 [7,8,9,10,11,12,13,14,15,16] and *Me* = 9 [5,6,7,8,9,10,11]). These differences were for PD_1 (“spontaneous exploration of the environment”; *p* = 0.00266) and PD_8 (“cognitive aptitude”; *p* = 0.00998; Appendix A). In both cases, children in the “No” category were significantly older (*Me* = 10 [7,8,9,10,11,12,13,14,15] and *Me* = 7 [5,6,7,8,9,10], and *Me* = 10 [7,8,9,10,11,12,13] and *Me* = 8 [5,6,7,8,9,10], respectively). After correcting for multiple comparisons, a significant difference was found only for PD_1 (“spontaneous exploration of the environment”; *p* = 0.03666). In the communication domain, only in CD_2 was the difference was significant (*p* = 0.02667), and the “No” children were significantly older (*Me* = 16 [11,12,13,14,15,16,17,18,19,20,21] and *Me* = 7 [5,6,7,8,9]; Appendix A). There were no age differences in ASD children in the SS, PD, or CD domains (*p* > 0.05; Appendix A).

When examining differences between children of different ages with DDX3X syndrome, there was a significant difference in the SS_5 (“strong desire to make friends”) category (*p* = 0.01664; Appendix A), and the “No” children were significantly older than the “Yes” children (*Me* = 10 [7,8,9,10,11,12,13,14,15,16] and *Me* = 6 [4,5,6,7,8,9,10], respectively). For the play domain, significant differences were found in the following three categories: PD_1 (“spontaneous exploration”; *p* = 0.00509), PD_8 (“cognitive aptitude”) (*p* = 0.01706), and PD_14 (“exceptional memory for details”) (*p* = 0.03555; Appendix A). The only difference when examining the age of children with *DDX3X* in the communication domain was found for CD_1 (“does your child speak”) (*p* = 0.01733; Appendix A).

To summarize, age in terms of skills was not a strong differentiating factor between the two groups.

## 4. Discussion

There is a need to differentiate ASD from other similar syndromes early on. Differences between ASD and other syndromes can be significant in therapy, daily life, or the prevention and treatment of comorbidities. 

An earlier study, and the only such one, comprising 23 female subjects, assessed the social and emotional functioning of girls and young women with *DDX3X* [15]. The results of our study are based on a group of 70 patients with *DDX3X* (4 males and 66 females), representing nearly 6% of the global population of this ultra-rare disease (https://ddx3x.org/, accession date: 7 August 2024). To date, positive social or communication traits of children with *DDX3X* have only been mentioned briefly in medical literature [15,16,27]. Based on the literature and our research, we have shown that social dysfunction is more characteristic of patients with ASD.

The findings of this study should help with adapting behavioral and educational therapy to the needs of people with *DDX3X* and contribute to more emphasis on using their strong, socially positive behaviors.

It has been shown that demographically there is a need for better diagnostics. Out of 21 patients from France tested with *DDX3X*, only two had a diagnosis of ASD. Funding for the WES trial in France is fully covered, which explains the low proportion of patients with a dual diagnosis. Parents are not looking for an additional diagnosis, as they receive the correct one straight away—*DDX3X*. The differences between ASD and *DDX3X* lie in competences, which can be misinterpreted and then over-diagnosed as autism. We describe these competences in the four areas that the questionnaire covered: medical, social skills, play, and communication.

Facial dysmorphia and microcephaly are factors that drive parents to seek an alternative diagnosis (Figure 2). However, not all patients manifest them, hence the need to identify other apparently autism-like characteristics that differentiate patients [31].

The medical literature commonly associates early gastrointestinal problems, including feeding and constipation in children, with *DDX3X* [5,15,32]. Gastrointestinal problems in *DDX3X* patients in the form of constipation were already identified by a Japanese study and are due to the DDX3X dysfunction characteristic of the *DDX3X* variant (NM_001193416.3: c.1574A > G; p.(Tyr525Cys)) [33]. Additionally, thirty-five de novo variants were identified in the *DDX3X* gene in females with intellectual disability, along with additional features such as hypotonia, movement disorders, behavioral problems, corpus callosum hypoplasia, and epilepsy [1]. The differences observed in our study, especially in the social skills and play domains, between ASD and *DDX3X*, could be a result of different gene variants. However, it is difficult to compare these observations, and we can only speculate, because we did not study genetic variants in patients.

Further medical characterization concerns the subjective experience of pain. Expression of pain in *DDX3X* patients may be related to the miR-181a-5p region in the *DDX3X* gene, with different sensations of pain or inflammatory transition in *DDX3X* patients [34]. There are studies that describe sensory abnormalities in patients with *DDX3X* [15,32]. Stereotyped movement disorders, hyperactivity, physical aggression, or self-injurious behaviors in children are mentioned as characteristics of children with *DDX3X* [1,5,13,15]. Children with *DDX3X* show high communicative intention [27].

No relationship between diagnosis type and being a “happy” and problem-free child in the first year of life was found in the questionnaire. Level of happiness has been measured in children with ASD [35]. This research has not yet been conducted on children with *DDX3X*.

Social traits appear to distinguish children with *DDX3X* from those with ASD. A total of 90% of parents of *DDX3X* patients declared that their child is sociable with friends, family, and people. Only 69% of ASD patients were assessed as such by their parents.

All participants in the Forbes study with *DDX3X* exhibited a sense of happiness. In contrast, in descriptions of children with *DDX3X* by other authors, parents mentioned humor and laughter as strengths of their daughters [15], but these characteristics were not identified in Ng Cordell’s research or ours [6,16]. They were able to use presymbolic communication intentionally (smiling when happy, for example), refusal, elicitation (requests, making choices), and social communication (in relation to greeting and saying farewell) [27]. This is confirmed by our questionnaire in the question about frequent smiling, laughing, and/or having a good sense of humor, as well as having a sense of happiness, which ranked at 97% in children with *DDX3X* and 68% in children with ASD. The opposite is true for children with ASD, i.e., even if a child is high-functioning in ASD (HFASD), they often have problems with social interactions, difficulties with peer relationships, and limited interests, which often leads to rejection and bullying [36,37,38,39,40].

The traits of having a good/friendly disposition, being friendly, and having a caring personality were rated differently in the two groups—in as many as 92% of children with *DDX3X* and in only 71% of children with ASD.

There is a brief mention of a caring and friendly personality in medical history interviews (MHIs) by parents of girls and young women with *DDX3X* [15] and also of a cheerful disposition and personality in MHIs by parents [16], and the validity of attributing these characteristics is confirmed by our study.

A powerful desire for friendship exhibited by their children was noted by up to 62% of parents of patients with *DDX3X*, and only by 16% of parents of children with ASD. In contrast, children with ASD have problems with this skill.

Similar findings were uncovered regarding the ability to develop and maintain friendships. An overwhelming number of parents of children with *DDX3X* (48%) answered in the affirmative, whereas only 23% in the ASD group did. High-functioning ASD (HFASD) patients have this ability; friendship follows a developmental trajectory in such patients, and they perceive friendship characteristics as similar to those of their friends and have the capacity for interpersonal awareness [41]. To date, these traits in children with ASD have only been mentioned twice. The first talked about a strong desire for friendships despite a lack of skills to develop and sustain them in MHIs by parents [15], and the second talked about a desire for social interaction [16].

Our results show a large difference in recognition of acquaintances and friends in the two groups (ASD = 76%, *DDX3X* = 100%, *p* = 0.00003). Deficits in facial emotion recognition [42,43,44], recognition of facial identity [45,46], face memory [47,48,49], face discrimination [50], face perception, and face recognition in everyday life [48] were previously reported in the scientific literature. Increased gaze towards the face/head and eyes is associated with higher social functioning in people with ASD. Abnormal visual processing interferes with social learning abilities of patients with ASD. The difficulties listed above comprise the primary deficit in ASD. These patients do not identify social partners as unique individuals and do not interpret facial expressions correctly, which results in difficulties with social interactions [46].

A publication with characteristics of 15 behavioral profiles showed elevated anxiety in 14% of subjects with *DDX3X* [32]. Levels of anxiety and self-harm were also identified in the *DDX3X* group [15]. However, intellectual disability (ID) was taken into account as a differentiating parameter. Interestingly, the level of autism traits was not different for the two groups [15]. Our questionnaire showed no association between anxiety-related behaviors (SS13, such as social withdrawal, fear of change in routine, or unusual anxiety) and diagnosis between the groups.

The part of the questionnaire devoted to play exhibited the most differences. There were significant differences in 8 of the 14 questions, while 2 exhibited a trend.

Natural environment teaching (NET) and incidental teaching (IT) is used in ABA therapy to encourage children to encounter reinforcing conditions in their environment so that they are motivated and more inclined to continue exploring their environment. The ability to spontaneously explore one’s environment must be intrinsically motivated. This spontaneity was described in a patient with *DDX3X* as a strong willingness to spontaneously explore the environment [16]. In the questionnaire, however, we did not identify an association between a diagnosis and the ability to explore the environment spontaneously. 

There is strong evidence that recreational swimming, competitive swimming and water aerobics can be an effective way to deal with mental health disorders [51]. A fascination with water in a patient with *DDX3X* that turned into an obsession was described in the literature [16]. In our study, more patients with *DDX3X* (91%) were fascinated by water than were those with ASD (76%), *p* = 0.023. Similar findings were uncovered for the love of swimming, where the majority of *DDX3X* patients loved to swim (94%), with far fewer ASD patients (78%), *p* = 0.01483 exhibiting this trait. Water therapy has been shown to be effective in children with ASD, e.g., in the aquatic speech and language therapy (ASLT) program [52]. Loving swimming was a trait often cited as a strength of *DDX3X* patients in MHIs with parents [15]. Our results may suggest the potential effectiveness of water-based therapy for children with *DDX3X* since the vast majority of this group loves to swim and is fascinated by water.

A total of 82% of *DDX3X* patients and only 64% of ASD patients loved to dance (*p* = 0.02611). Recent studies on dance have shown positive effects on cognitive function (memory, attention, body balance, psychosocial parameters) and improved peripheral neurotrophic factors [53], as well as positive effects on motivation and stress reduction [54]. Dancing is a physical activity that is proven to provide physiological and psychosocial benefits for people with neurodevelopmental disabilities [55]. Love of dancing was a trait mentioned in MHIs by parents of patients with *DDX3X* [15]. Dance therapy has not yet been studied in children with *DDX3X*, but this activity may have great therapeutic potential in this patient group.

In contrast, spending time together with family, playing, and dancing with other people was enjoyed by more *DDX3X* patients (93%) and by significantly fewer ASD patients (74%) in our study *p* = 0.00604. The observation that children with ASD often come into contact with an object and rarely engage in social play has been noted by Kanner [56]. Lack of mutual attention makes it difficult for children with ASD to learn ways to interact with objects and people around them [57]. Children with ASD show less playfulness and joy during play [58].

Furthermore, in our study, children with *DDX3X* willingly joined in play with other children, even despite not knowing the rules of the game (as much as 70%; *p* = 0.00015). This figure was as low as 33% in the group of ASD children. One of the characteristics of play is the intrinsic motivation of children to play, their desire to participate of their own free will, and action as a process that is more important to them than the outcome [59,60]. The participants in our study exhibited these characteristics. Our results suggest the existence of a mutual attentional field in children with *DDX3X*, so future assessment of play by children with *DDX3X* should also take into account the shared attentional field.

Moreover, children with *DDX3X* (81%) were more likely to initiate spontaneous conversations with a parent than children with ASD (60%) (*p* = 0.0112). It has been proven that the absence or deficiency of symbolic play can serve as an early marker of ASD. Children with ASD exhibit more difficulties in symbolic play than children with other neurodevelopmental and other language disorders (DLDs) at comparable developmental levels [61]. Children with ASD are less likely to engage in such play [62]. This is particularly evident in free or spontaneous play [59]. Symbolic play in children with ASD is learned, and although it may be shaped, it lacks spontaneity and novelty [63]. Autistic children do not have difficulty in imitating symbolic play but rather in generating it, which is linked to lack of imagination [64]. Lack of spontaneous communication initiation appears to be a persistent problem for people with ASD. Spontaneous communication is less frequent compared to people with language disorders and typically developing people. Furthermore, 21–66% of children with ASD do not develop communication skills. People with ASD rarely initiate appropriate verbal communication and often do not engage in typical social interactions such as asking questions, requesting information, expressing feelings, or asking for interaction [65]. Our results may suggest that patients with *DDX3X* manifest spontaneous play, as opposed to patients with ASD.

The traits in which ASD patients excelled include exceptional memory for details (88%), which *DDX3X* patients manifested significantly less frequently (64%), *p* = 0.00588. 

In terms of cognitive aptitude, we did not identify a relationship between cognitive aptitude and diagnosis. One child with *DDX3X* is described in the literature as exhibiting cognitive curiosity [16]. Extraordinary talents (ETs) are, as recent research has shown, common among children with ASD [66,67]. Recent findings suggest that with age, adaptive functioning deteriorates in ASD patients, and this also occurs among intellectually gifted children with ASD; their cognitive ability is not a protective factor [68]. To date, a child with *DDX3X* who was moved to an inclusive educational facility attended by children with a range of disabilities, not just ASD, and has made progress is only mentioned in one paper [16]. The results of this questionnaire in the play category clearly confirm significant differences in this area between children with *DDX3X* and ASD.

There were no statistical differences in questionnaire responses in the communication category between the *DDX3X* and ASD groups. We did not identify an association between a diagnosis and the child’s ability to speak, use of functional communication, problems in expressing needs and at the same time wanting to be understood, understanding speech, or pointing a finger at an object out of reach. Both children with ASD and those with *DDX3X* manifest speech and communication difficulties. However, speech is not the only form of communication. Patients can manifest functional communication through eye contact and the use of gestures.

Our questionnaire identified a difference in one question, which asked whether the patient was checking their carer’s reaction to a new situation in their surroundings—for example, when a new sound or a person not seen for a long time appears (*p* = 0.01917).

Communicating via speech was shown to be problematic for patients with DDX3X syndrome. Children with *DDX3X* use other ways to communicate, smiling when they are happy, gesturing to indicate refusal, and using social communication to say hello and goodbye. Communication and social functioning needs have been separated from speech [27]. Patients with even severe speech and language impairment exhibited social motivation. This is because damage to the central nervous system is different in patients with ASD and those with *DDX3X* [27].

### Limitations, Strengths, and Future Directions

Sample size is the only weakness of our study; however, DDX3X syndrome is an ultra-rare disease. This is the largest study using a newly developed social behavior questionnaire involving both boys and girls in a test and a control group. The ages of the participants were similar, which is a strength of this study. We are planning to continue research in the future, hoping to establish new contacts, and thus expand the sample size, at future DDX3X syndrome-dedicated conferences.

We designed the questions based on our own observations of free play of children with *DDX3X* at the 7th Annual *DDX3X* Foundation Scientific Conference, held between 11 and 13 November 2021; and the 1st International Conference on DDX3X Syndrome in Paris, held on 20 October 2023; as well as the available medical literature. Not every question proved the validity of our own observations, so in future we would like to ask parents to participate in research again on a larger scale, using the already standardized scale described in this article. Future research should focus on play in children with *DDX3X*, using the children’s peers as models. In addition, research is needed to assess the impact of the exposure of children with *DDX3X* to playing with peers. It is important to check if it has an impact on language development and how this type of play affects other areas of a child’s development. In another article, we would like to describe how we designed the questionnaire and how we selected the questions. In another article, we would like to present the perspective of a parent of a child with *DDX3X* from a knowledge base that parents provided us with in the additional comments section to understand the perspective of raising a child with *DDX3X*.

## 5. Conclusions

Some of the traits common in *DDX3X* overlap with those in ASD and are attributed to autism. This questionnaire charts a new, innovative approach for diagnoses differentiating autism from *DDX3X*. Our questionnaire is the first study that unequivocally proves that, in contrast to ASD patients, 100% of DDX3X syndrome respondents recognize their friends and acquaintances. This trait, which is a basic ASD deficit, has been shown to be a strength for children with *DDX3X*, which should be exploited in therapy. This is a very important marker for further differential diagnosis of patients. 

Differences between disorders are the basis for designing better educational and therapeutic approaches. Educational therapy directed at social skills that children with *DDX3X* have and the types of play they prefer can contribute to better social functioning. Patients’ communication presents a diagnostic problem, and thus differentiating *DDX3X* from ASD should be taken into account in patients’ clinical observations and family history. A test can be used routinely to screen for *DDX3X* in children who present language development delays and developmental problems, and where symptoms do not correspond to typical traits of ASD. If therapeutic interventions for patients with *DDX3X* are to be effective, it is important for therapists to have an appropriate assessment of the patients’ social skills.

There is a powerful need to develop concise and sensitive research tools that will provide a clue in capturing a profile of a child with *DDX3X*. The questionnaire can become a quick and easy-to-use tool for parents, who will be able to capture the profile of a child with DDX3X on this basis. Improved detection can in turn contribute to better diagnoses of this ultra-rare disease. The new findings of this study may have important research and clinical implications, as WES in some countries is costly and parents forgo further diagnosis. Our questionnaire is cost-free and may convince parents to investigate further.

An early diagnosis helps avoid a diagnostic odyssey and can help win some time for early intervention. Our questionnaire emphasizes the importance of play and social skills. These findings may facilitate better identification of individuals at risk of *DDX3X* and the development of effective interventions to help children with *DDX3X* worldwide.

Further research on a larger number of patients may bring to light more differences in this area, the possibility of faster and more accurate diagnosis, and a move away from the diffuse phenotypic concept of the “*spectrum*” [22].

## Figures and Tables

**Figure 1 jcm-13-07842-f001:**
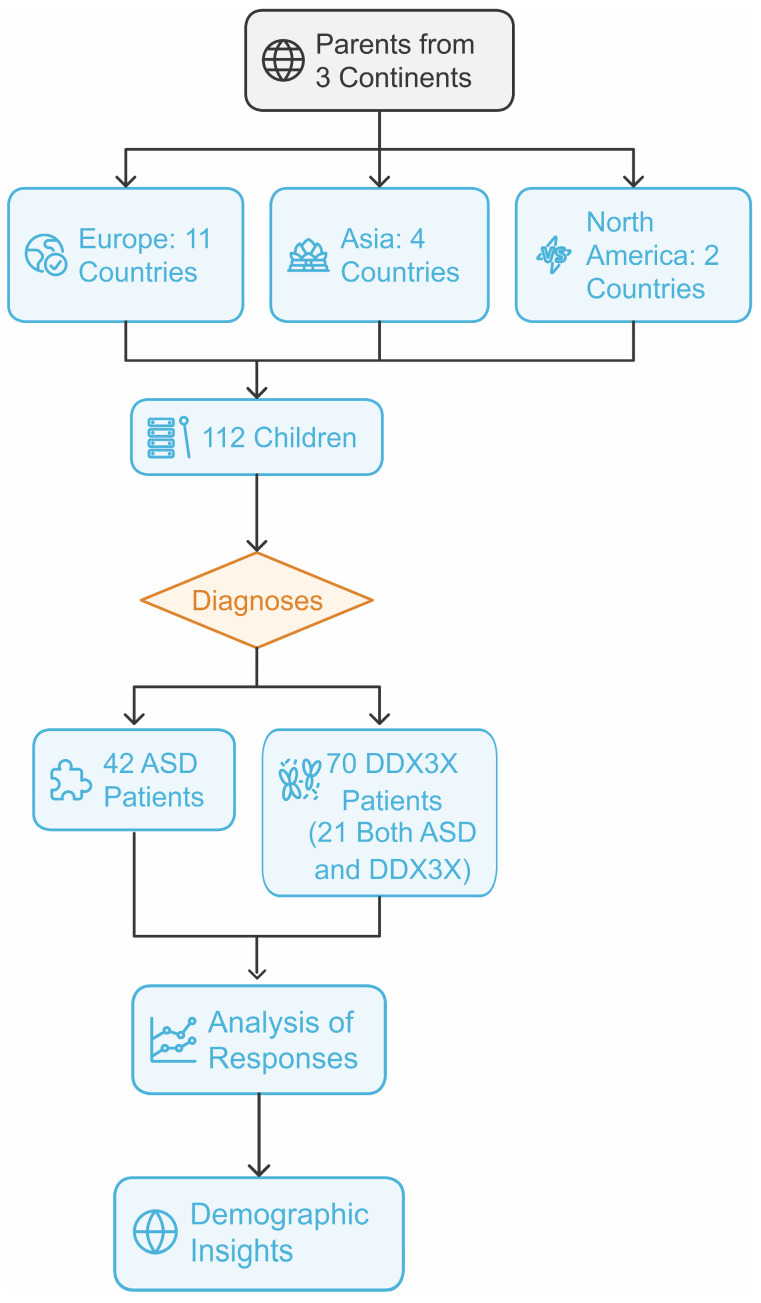
This flowchart outlines the study design. Parents from three continents (Europe: 11 countries; Asia: 4 countries; North America: 2 countries) contributed to the sample of 112 children. These children were diagnosed, resulting in 42 patients with ASD and 70 patients with DDX3X syndrome (with 21 patients with both ASD and DDX3X).

**Figure 2 jcm-13-07842-f002:**
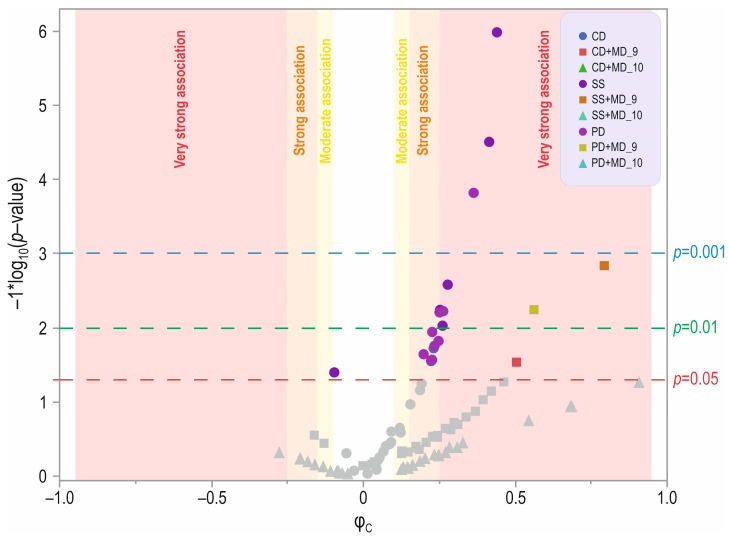
Volcano plot showing statistical significance and magnitude of change in the analyzed domains between children with autism spectrum disorder (cases with ASD, DDX3X syndrome negative) and children with DDX3X syndrome (children with DDX3X syndrome without and with ASD). Y-axis—increasing significance level (showed as −1 × log_10_), X-axis—magnitude of association with Cramér’s phi measure (φ_c_). Designations: CD—communication domain, SS—social skills domain, PD—play domain, MD—medical domain (9—facial dysmorphia, 10—microcephaly). The grey symbols represent insignificant differences.

**Figure 3 jcm-13-07842-f003:**
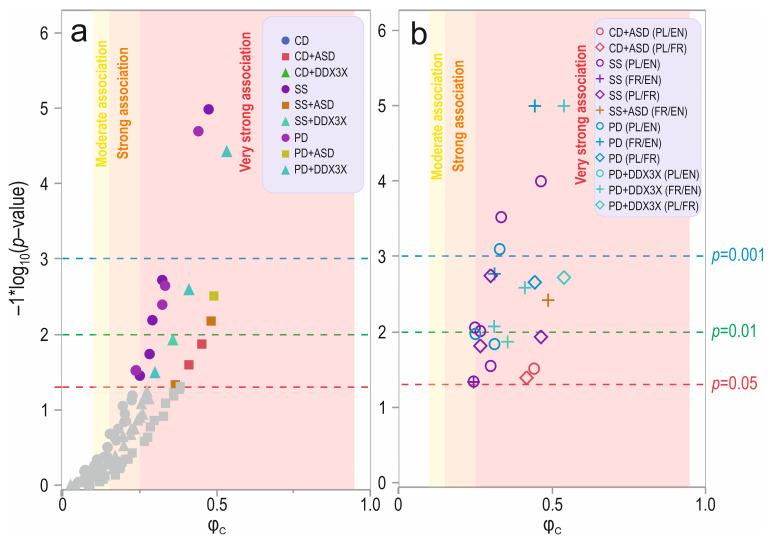
Volcano plot showing statistical significance and magnitude of change between (**a**) children speaking different languages in the analyzed domains and (**b**) between specific languages (Polish [PL], French [FR], and English [EN]) in the analyzed domains. Y axis—increasing significance level (showed as −1 × log_10_), X axis—magnitude of the association with Cramér’s phi measure (φ_c_). Designation: ASD—children affected by autism spectrum disorders (cases with ASD, DDX3X syndrome negative), DDX3X syndrome—children with DDX3X syndrome without or with ASD. Designations: CD—communication domain, SS—social skills domain, PD—play domain. The grey symbols represent insignificant differences.

**Figure 4 jcm-13-07842-f004:**
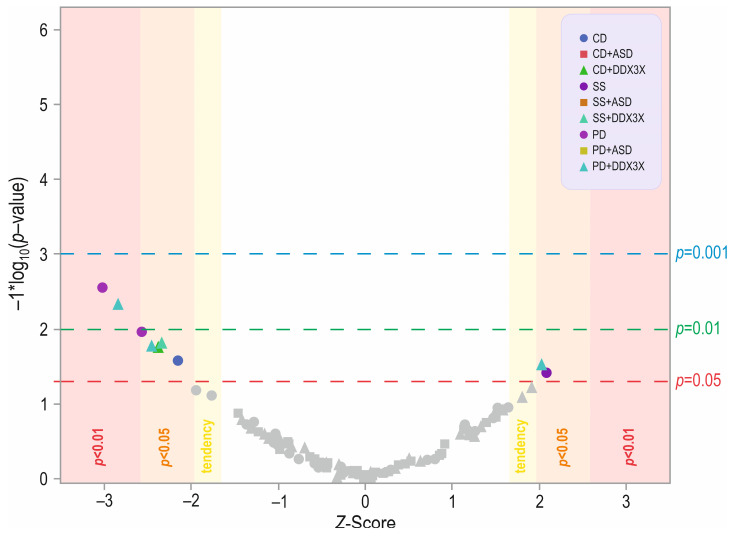
Volcano plot showing statistical significance and magnitude of change in age of children between the analyzed groups. Y axis—increasing significance level (showed as −1 × log_10_), X axis—magnitude of association with *Z*-score level. The trend could be interpreted for between *p* = 0.01 and *p* = 0.05. Designation: ASD—children affected by autism spectrum disorders (cases with ASD, DDX3X syndrome negative), DDX3X syndrome—children with DDX3X syndrome without or with ASD. Designations: CD—communication domain, SS—social skills domain, PD—play domain. The grey symbols represent insignificant differences.

## Data Availability

The anonymized dataset used and/or analyzed during the current study is available from the corresponding authors on reasonable request.

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
