# Peer review of "First Diagnostic Questionnaire for Assessing Patients’ Social Functioning: Comprehensive DDX3X Syndrome Patient Profile"

_jcm, 2024, doi:10.3390/jcm13247842_

Round 1

Reviewer 1 Report

Comments and Suggestions for Authors

In the article “Comprehensive DDX3X Syndrome Patient Profile: First Diagnostic Questionnaire for Assessing Patient Functioning”, the authors used questionnaires on items such as social skills, play and communication to look for differences between patients with ASD and DDX3X syndrome. This article describes the results of an interesting and important study.

The list of references is sufficient and relevant.

Overall, the paper is well written, but there are some comments..

It may be worth tweaking the title, as the phrase ‘Patient Functioning’ is vague and could refer to any aspect of patients' lives.

Materials and methods are described in a clear and correct manner.

The main comment relates to the lack of comparison with children without intellectual disabilities. The results for most of the items investigated will be close between the groups of children with ASD and DDX3X syndrome. However, the results for some items in children with DDX3X syndrome may be closer to children without intellectual disabilities. Understanding the degree of deviation from the behaviour of healthy children can be of diagnostic value.

It may be worth changing the type of graphs used to visualise the results. This is especially true for Figure 2. The figure does not reflect the groups of children by language, so the meaning is lost.

Author Response

Response to Reviewer 1 Comments

1. Summary

2. Questions for General Evaluation

Reviewer’s Evaluation

Response and Revisions

Does the introduction provide sufficient background and include all relevant references?

Yes

Thank You for positive evaluation

Is the research design appropriate?

Can be improved

Thank You for evaluation, we will improve manuscript with suggested amendments

Are the methods adequately described?

Yes

Thank You for positive evaluation

Are the results clearly presented?

Can be improved

Thank You for evaluation, we will follow suggested amendments

Are the conclusions supported by the results?

Yes

Thank You for positive evaluation

3. Point-by-point response to Comments and Suggestions for Authors

Comments 1: In the article “Comprehensive DDX3X Syndrome Patient Profile: First Diagnostic Questionnaire for Assessing Patient Functioning”, the authors used questionnaires on items such as social skills, play and communication to look for differences between patients with ASD and DDX3X syndrome. This article describes the results of an interesting and important study.

Response 1: Thank You for pointing this out and for highlighting how important and interesting our study is.

Comments 2: The list of references is sufficient and relevant.

Response 2: Agree. Thank You for Your positive evaluation.

Comments 3: Overall, the paper is well written, but there are some comments.

It may be worth tweaking the title, as the phrase ‘Patient Functioning’ is vague and could refer to any aspect of patients' lives.

Response 3: Thank You for paying attention to the title. We clarified the title of the papers to the aspect of social functioning.

Comments 4: Materials and methods are described in a clear and correct manner.

Response 4: Agree. Thank You for Your positive evaluation.

Comment 5: The main comment relates to the lack of comparison with children without intellectual disabilities. The results for most of the items investigated will be close between the groups of children with ASD and DDX3X syndrome. However, the results for some items in children with DDX3X syndrome may be closer to children without intellectual disabilities. Understanding the degree of deviation from the behaviour of healthy children can be of diagnostic value.

Response 5: Thank You for Your suggestion. Please see our clarification in the manuscript (lines 99-102).

Comment 6: It may be worth changing the type of graphs used to visualise the results. This is especially true for Figure 2. The figure does not reflect the groups of children by language, so the meaning is lost.

Response 6: We appreciate the reviewer's suggestion. Given the large number of comparisons, the volcano-plots seems to be most reasonable in our opinion. We agree with the reviewer, accordingly the language-dependent differences, thus we have carefully considered alternative visualization for Figure 2. Ultimately, we have opted for two-panel figure (a) remain the same and the second one (b) shows only significant sub-groups’ differences, as it effectively conveys the following key information.

4. Response to Comments on the Quality of English Language

Point 1: The quality of English does not limit my understanding of the research.

Response 1: Thank You for Your positive evaluation.

Yours faithfully,

Urszula Stefaniak-Preis

Reviewer 2 Report

Comments and Suggestions for Authors

Summary
The authors presented an intriguing analysis of the first diagnostic questionnaire designed for assessing patient functioning in individuals with DDX3X syndrome. While the work is compelling, certain limitations noted in the Discussion section must be addressed. Furthermore, incorporating a statistical model to define the optimal sample size needed for validating the study would strengthen its conclusions.

Title
Since the primary focus of your work is the questionnaire, it would be beneficial to restructure the title to highlight the questionnaire first, followed by its application in profiling DDX3X syndrome patients.

Authors
It is unnecessary to include ORCID information in the manuscript itself. However, ensure that the listed ORCIDs are confirmed and active on the ORCID website.

Abstract

  • Line 24: Provide a brief mention of the study's aim and experimental setup before introducing the questionnaire. This adjustment will give readers a clearer context for the work.
  • Lines 30–31: Refine the final lines of the abstract to emphasize the significance and broader implications of your study, providing a strong takeaway message for readers.

Discussion

  • Include additional details about the genetic variants of DDX3X, as briefly mentioned in lines 336–337.
  • Discuss the annotations available for the DDX3X gene in the OMIM database. Compare the clinical synopsis in OMIM (https://omim.org/entry/300160?search=ddx3x&highlight=ddx3x) with the phenotypic descriptions in your study, highlighting similarities and differences.
  • Line 491: Given the limitations associated with sample size, consider providing a statistical model to estimate the appropriate sample size needed to achieve significant p-values in your cohort. This addition would help address concerns related to sample size constraints, especially given the ultra-rare nature of the variants studied.

Author Response

For research article

Response to Reviewer 2 Comments

1. Summary

Thank You very much for taking the time to review this manuscript. Please find the detailed responses below and the corresponding revisions/corrections highlighted/in track changes in the re-submitted files.

2. Questions for General Evaluation

Reviewer’s Evaluation

Response and Revisions

Does the introduction provide sufficient background and include all relevant references?

Yes

Thank You for positive evaluation

Is the research design appropriate?

Can be improved

Thank You for evaluation, we will fallow suggested amendments

Are the methods adequately described?

Can be improved

Thank You for evaluation, we will follow suggested amendments

Are the results clearly presented?

Can be improved

Thank You for evaluation, we will follow suggested amendments

Are the conclusions supported by the results?

Can be improved

Thank You for your evaluation, we will follow suggested amendments

3. Point-by-point response to Comments and Suggestions for Authors

Comments 1: Summary. The authors presented an intriguing analysis of the first diagnostic questionnaire designed for assessing patient functioning in individuals with DDX3X syndrome.

Response 1: Thank You for Your positive evaluation and pointing this out.

Comments 2: While the work is compelling, certain limitations noted in the Discussion section must be addressed. Furthermore, incorporating a statistical model to define the optimal sample size needed for validating the study would strengthen its conclusions.

Response 2: Thank You for Your suggestion. Please see Response 6, 7 and 8.

Comments 3: Title. Since the primary focus of your work is the questionnaire, it would be beneficial to restructure the title to highlight the questionnaire first, followed by its application in profiling DDX3X syndrome patients.

Response 3: Thank You very much for Your suggestion. We rearranged and changed the title as follows: First Diagnostic Questionnaire for Assessing Patients' Social Functioning: Comprehensive DDX3X Syndrome Patient Profile.

Comments 4: Authors. It is unnecessary to include ORCID information in the manuscript itself. However, ensure that the listed ORCIDs are confirmed and active on the ORCID website.

Response 4: Thank You very much for pointing this out. As suggested we removed the ORCID information in the manuscript body, and checked the accuracy in the submission system.

Comment 5: Abstract.

  • Line 24: Provide a brief mention of the study's aim and experimental setup before introducing the questionnaire. This adjustment will give readers a clearer context for the work.
  • Lines 30–31: Refine the final lines of the abstract to emphasize the significance and broader implications of your study, providing a strong takeaway message for readers.

Response 5:

Line 24: We thank the reviewer for this amendment. We agree that providing a brief overview of the study's aim and experimental design before introducing the questionnaire will enhance the clarity of the manuscript. Please see the lines 25-28 of the manuscript.

Lines 30-31:

As suggested, we changed the last sentence of the abstract, to emphasize the results of our study as follows: "Identification of early behavioral markers that differentiate children with ASD and DDX3X could lead to earliest opportunity to identification and intervention, and can significantly impact developmental trajectories, leading to better long-term outcomes." Please see the lines 34-28 of the manuscript.

Comment 6: Discussion. Include additional details about the genetic variants of DDX3X, as briefly mentioned in lines 336–337.

Response 6: We included information regarding thirty-five de novo variants in the DDX3X gene in females with intellectual disability and additional features such as hypotonia, movement disorders, behavioral problems, corpus callosum hypoplasia, and epilepsy. Please see the lines 386-392 in the manuscript.

Comment 7: Discuss the annotations available for the DDX3X gene in the OMIM database. Compare the clinical synopsis in OMIM (https://omim.org/entry/300160?search=ddx3x&highlight=ddx3x) with the phenotypic descriptions in your study, highlighting similarities and differences.

Response 7: Thank You for Your suggestion. As requested, we included some additional information regarding the commonly known 35 variants of DDX3X in an appropriate place. Please see the lines 386-392.

Comment 8: Line 491: Given the limitations associated with sample size, consider providing a statistical model to estimate the appropriate sample size needed to achieve significant p-values in your cohort. This addition would help address concerns related to sample size constraints, especially given the ultra-rare nature of the variants studied.

Response 8:

We thank the reviewer for their insightful comment regarding the sample size limitations. We acknowledge that the ultra-rare nature of the variants studied presents challenges in achieving statistical significance. The study's sample size was restricted by the population of children meeting inclusion criteria (DDX3X), thereby limiting the statistical power and generalizability of the results. The statistical power was determined posteriori following data collection. As the fractions of ASD and DDX3X children were 62.5% to 37.5% respectively, the achieved statistical power was 0.73 for the group of 112 children. We added the information in the results section. Additionally, we are planning to continue our research in the future, hoping to establish new contacts and thus expand the sample size at future DDX3X-syndrome-dedicated conferences. We are also in a process of organizing 2nd International DDX3X Conference at Poznan University of Medical Sciences on 26th March 2025 in Poland.

4. Response to Comments on the Quality of English Language

Point 1: The quality of English does not limit my understanding of the research.

Response 1: Agree. Thank You for Your positive evaluation.

Yours faithfully,

Urszula Stefaniak-Preis

Reviewer 3 Report

Comments and Suggestions for Authors

The research titled "Comprehensive DDX3X Syndrome Patient Profile: First Diagnostic Questionnaire for Assessing Patient Functioning" by Stefaniak-Preis et al. is one of the interesting research articles. This article describes the DDX3X Syndrome.

Merits:

The introduction is well-written.

The research study is well-designed.

The authors have used a reasonable sample size for this study from various regions.

However, the authors should address the following queries to improve the manuscript.

  1. The abstract could be more precise. It mainly describes the introduction, and the original findings are expressed only in a few lines. The authors should work with the community to present the original findings in the abstract.
  2. Presenting the study design as a scheme would be more beneficial to this manuscript, and also presenting the participant details
  3. The authors do not mention their overall findings at the end of each category's results section.
  4. All the figures are blurred and should be clearly presented

Author Response

For research article

Response to Reviewer 3 Comments

1. Summary

2. Questions for General Evaluation

Reviewer’s Evaluation

Response and Revisions

Does the introduction provide sufficient background and include all relevant references?

Yes

Thank You for Your positive evaluation

Are the methods adequately described?

Can be improved

Thank You for Your evaluation. We will follow suggested amendments

Are the results clearly presented?

Must be improved

Thank You for Your evaluation. We will follow suggested amendments

Are the conclusions supported by the results?

Can be improved

Thank You for Your evaluation. We will follow suggested amendments

3. Point-by-point response to Comments and Suggestions for Authors

Comments 1: The research titled "Comprehensive DDX3X Syndrome Patient Profile: First Diagnostic Questionnaire for Assessing Patient Functioning" by Stefaniak-Preis et al. is one of the interesting research articles.

Response 1: Thank You for pointing this out. We agree with this comment. Thank you for your positive evaluation.

Comments 2: This article describes the DDX3X Syndrome.

Merits: The introduction is well-written.

The research study is well-designed.

The authors have used a reasonable sample size for this study from various regions.

Response 2: Agree. We agree with the above comments. Thank you for your positive evaluation.

Comments 3: However, the authors should address the following queries to improve the manuscript. The abstract could be more precise. It mainly describes the introduction, and the original findings are expressed only in a few lines. The authors should work with the community to present the original findings in the abstract.

Response 3: Thank You for the suggestion. Considering the abstract word count limitations, we could only adapt to this recommendation to a limited extent.

Comments 4: Presenting the study design as a scheme would be more beneficial to this manuscript, and also presenting the participant details.

Response 4: Thank You for Your suggestion. A flow-chart diagram was added to present details. Please see the flow chart on page 4 of the corrected manuscript.

Comments 5: The authors do not mention their overall findings at the end of each category's results section.

Response 5: Thank You for Your precious suggestion. We have added the summarize after each of category’s results section:

Line 229-231

Line 247-249

Line 307-308

Line 346-347

Comments 6: All the figures are blurred and should be clearly presented. 

Response 6:  Thank You for Your suggestion. All the figures were exchanged for better-quality images. Additionally, Figure 2 was changed according to another reviewer's suggestion. Additionally, if the paper is accepted, inform the technical team of MDPI to take care of this issue.

4. Response to Comments on the Quality of English Language

Point 1: The quality of English does not limit my understanding of the research.

Response 1: Agree. Thank You for Your positive evaluation.

Faithfully Yours,

Urszula Stefaniak-Preis

Reviewer 4 Report

Comments and Suggestions for Authors

Review

The scope of the current study was to investigate the behavioural differences between individuals with ASD and XXD3X syndrome using a new tool, namely the Social Behavior Questionnaire. The aim of the study is relevant to the Journal and has great merit for the clinicians, who work with children with neurodevelopmental disorders.

General comments

Even though the authors declare their desire to explain more about the development of the Social Behavior Questionnaire in a future manuscript, I strongly believe that is essential to provide more information about the new tool in this paper, especially as it concerns the way of scoring and the methodology used to manipulate possible uncompleted questionnaires, as these issues in methodology can influence significantly the results presented.

Introduction

Line 63. A citation is missing after the global prevalence of ASD.

Line 57. Please, explain what RD in the DDX3X-RD stands for before you use it.

Line 63-64. You are referring that individuals with ASD do not have access to a comprehensive genetic test. Please, explain the reasons you consider this necessary.

Line 105. I suppose it is DDX3X. Please correct it.

Material and Methods

Lines 94-102. The authors should describe the Social Behavior Questionnaire in more detail. How many items are included in each domain of the tool? Which items do not take a mark and why? Is there a unifying way of scoring? Which is that? All the questionnaires collected were fully completed? What did the authors do with the uncompleted questionnaires?

Results

Lines 131-132. The findings about the reliability of the tool should be transferred to the results section.

Lines 135-136. From which analyses have the referred questions been excluded?

Line 274. These sentences do not make sense. Something is missing. Please, correct them.

Discussion

Line 324. Authors should explain what the abbreviation WES stands for before they use it.  

Line 455. “of children z ASD” Please correct the typing error.

Author Response

For research article

Response to Reviewer 4 Comments

1. Summary

Thank You very much for taking the time to review this manuscript. Please find the detailed responses below and the corresponding revisions/corrections highlighted/in track changes in the re-submitted files.

2. Questions for General Evaluation

Reviewer’s Evaluation

Response and Revisions

Does the introduction provide sufficient background and include all relevant references?

Yes

Thank You for Your positive evaluation

Is the research design appropriate?

Yes

Thank You for Your positive evaluation

Are the methods adequately described?

Can be improved

Thank you for Your evaluation, we will follow the suggested amendments

Are the results clearly presented?

Yes

Thank You for Your positive evaluation

Are the conclusions supported by the results?

Yes

Thank You for Your positive evaluation

3. Point-by-point response to Comments and Suggestions for Authors

Comments 1: The scope of the current study was to investigate the behavioural differences between individuals with ASD and XXD3X syndrome using a new tool, namely the Social Behavior Questionnaire. The aim of the study is relevant to the Journal and has great merit for the clinicians, who work with children with neurodevelopmental disorders.

Response 1: Thank You for pointing this out. We agree with this comment.

Comments 2: Even though the authors declare their desire to explain more about the development of the Social Behavior Questionnaire in a future manuscript, I strongly believe that is essential to provide more information about the new tool in this paper, especially as it concerns the way of scoring and the methodology used to manipulate possible uncompleted questionnaires, as these issues in methodology can influence significantly the results presented.

Response 2: Thank You for Your precious suggestion. The social Behavior Questionnaire is the domain trully differ children with ASD and children with DDX3X. That is why we provided more information in the article in the lines 117-126.

Comment 3: Line 63. A citation is missing after the global prevalence of ASD.

Response 3: Thank You for noticing lack of citation. The missing reference was added. Please see the line 70.

Comment 4: Line 57. Please, explain what RD in the DDX3X-RD stands for before you use it.

Response 4: Thank You for pointing this out. We have explained what RD in the DDX3X-RD stands for. Please see line 64.

Comment 5: Line 63-64. You are referring that individuals with ASD do not have access to a comprehensive genetic test. Please, explain the reasons you consider this necessary.

Response 5: Thank you for this important comment. The explanation has been added to the manuscript in lines 71-73.  

Comment 6:  Line 105. I suppose it is DDX3X. Please correct it.

Response 6: Thank You for pointing this typing error. We have corrected it, please see line 132.

Comment 7: Lines 94-102. The authors should describe the Social Behavior Questionnaire in more detail. How many items are included in each domain of the tool? Which items do not take a mark and why? Is there a unifying way of scoring? Which is that? All the questionnaires collected were fully completed? What did the authors do with the uncompleted questionnaires?

Response 7: Thank You for pointing this suggestions. We have added the above to the manuscript. Please see the lines 117-126 in the corrected manuscript.

Comment 8: Lines 131-132. The findings about the reliability of the tool should be transferred to the results section.

Response 8: Thank You for pointing this out. We have transferred the results about the reliability of the tool to the results section. Please see the line 173.

Comment 9: Lines 135-136. From which analyses have the referred questions been excluded?

Response 9: Thank You for pointing the word missing, we ment to write statistical analysis. This has been added to the manuscript. Please see the line 168 in corrected manuscript.

Comment 10: Line 274. These sentences do not make sense. Something is missing. Please, correct them.

Response 10: Thank you for pointing this out. The sentence was corrected.

Comment 11: Line 324. Authors should explain what the abbreviation WES stands for before they use it.  

Response 11: Thank You for your comment. The abbreviation WES has been explained in the line 65 when it first appear in the manuscript.

Comment 12: Line 455. “of children z ASD” Please correct the typing error.

Response 12: Thank You for pointing this typing error. The error has been corrected, please see the line 510.

4. Response to Comments on the Quality of English Language

Point 1: The quality of English does not limit my understanding of the research.

Response 1: Agree. Thank You for Your positive evaluation.

Yours faithfylly,

Urszula Stefaniak-Preis

Round 2

Reviewer 2 Report

Comments and Suggestions for Authors

Authors addressed all the Reviewer's Comments